**∂ | Open Peer Review** | Clinical Microbiology | Research Article

# Carbapenem-resistant *Acinetobacter baumannii* at a hospital in Botswana: detecting a protracted outbreak using whole genome sequencing

Jonathan Strysko,[1,2,3,4] Tefelo Thela,[5] Andries Feder,[2] Janet Thubuka,[5] Tichaona Machiya,[5] Jack Mkubwa,[5] Kagiso Mochankana,[5] Celda Tiroyakgosi,[5] Kgomotso Kgomanyane,[5] Tlhalefo Dudu Ntereke,[3] Tshiamo Zankere,[3] Kwana Lechiile,[3,6] Teresia Gatonye,[3,7] Chimwemwe Viola Tembo,[3] Moses Vurayai,[7] Naledi Mannathoko,[7] Margaret Mokomane,[6] Ahmed M. Moustafa,[2,8] David M. Goldfarb,[8] Melissa Richard-Greenblatt,[9,10] Carolyn McGann,[2,4] Susan E. Coffin,[2,4] Britt Nakstad,[1] Corrado Cancedda,[3,4] Ebbing Lautenbach,[8] Dineo Bogoshi,[11] Anthony M. Smith,[11,12] Paul J. Planet[2,4]

**ABSTRACT** Carbapenem-resistant *Acinetobacter baumannii* (CRAb) has emerged as a major and often fatal cause of bloodstream infections among hospitalized patients in low- and middle-income countries (LMICs). CRAb outbreaks are hypothesized to arise from reservoirs in the hospital environment, but outbreak investigations in LMICs are often limited in scope due to lack of access to whole genome sequencing (WGS). We performed WGS on 43 stored isolates (blood cultures [$n = 23$] and environmental swabs [$n = 20$]) presumptively identified as *A. baumannii* collected during 2021–2022 from a 530-bed referral hospital in Gaborone, Botswana, where CRAb infection incidence was rising. Taxonomic assignment, multilocus sequence typing, antimicrobial resistance gene identification, K and O locus typing, and phylogenetic analyses were performed using publicly accessible analysis pipelines. All 23 blood and 25% (5/20) of environmental isolates were confirmed as *A. baumannii,* 79% ($n = 22$) of which were sequence type 1 (ST1). All ST1 isolates harbored genes encoding carbapenemases ($bla_{NDM-1}$, $bla_{OXA-23}$). Phylogenetic analysis demonstrated that nearly identical ST1 isolates spanned wide ranges in time (>1 year), suggesting ongoing transmission from environmental sources. One highly similar clade (average difference of 2.3 single nucleotide polymorphisms) contained all eight neonatal blood isolates and three environmental isolates from the neonatal unit. Environmental isolates included a sample from a sink drain, which is likely a major reservoir in this setting and highlights the need for targeted environmental remediation. Using a phylogenetically informed approach, we also identified diagnostic genes that distinguish this outbreak clone. These markers hold the potential to provide a low-cost method for tracking future CRAb isolates related to this outbreak.

**IMPORTANCE** Carbapenem-resistant *Acinetobacter baumannii* is an increasingly significant cause of hospital-acquired bloodstream infections, particularly in low- and middle-income countries where limited resources often prevent the use of advanced outbreak investigation methods. This study leveraged whole genome sequencing to uncover transmission patterns of these antibiotic-resistant infections which were occurring more frequently in a referral hospital in Botswana. By linking clinical and environmental samples collected over an 18-month period, we identified a cluster of infections genetically linked to samples collected from the environment, including a sample taken from a sink drain in the neonatal unit. Furthermore, the study identified key genes specific to outbreak strains that could be used as diagnostic markers to track future outbreaks, even in the absence of genomic sequencing. These findings demonstrate how combining genomic sequencing with targeted gene identification can

Address correspondence to Jonathan Strysko, jstrysko@gmail.com.

The authors declare no conflict of interest.

guide infection prevention and control efforts, helping to curb the spread of antibiotic resistance in resource-limited settings.

**KEYWORDS** *Acinetobacter*, hospital infections, carbapenems, environmental microbiology, genome analysis, infectious clones

Antimicrobial resistance (AMR) disproportionately affects low- and middle-income countries (LMICs), with sub-Saharan Africa bearing the highest burden (1). Carbapenem antibiotics are considered last-line agents for infections due to multidrug-resistant gram-negative organisms, but the global prevalence of carbapenem resistance is now rapidly increasing (2). In 2019, carbapenem-resistant *Acinetobacter baumannii* (CRAb) was estimated to be the fourth leading pathogen responsible for AMR-attributable mortality and the World Health Organization named it a "priority-one pathogen" for research and development of new antibiotics (1, 3). Surveillance data from LMICs demonstrate a pattern of CRAb disproportionately affecting neonates and infants; CRAb is now among the top five etiologies of neonatal BSI in many countries in sub-Saharan Africa and South Asia (4–8). At the tertiary referral hospital in Botswana where this study was carried out, bloodstream infection surveillance records indicated that between 2012 and 2021, the proportion of *A. baumannii*-attributable neonatal sepsis rose from 1% to 16% and was associated with a case fatality rate of 56% (9, 10).

CRAb has been closely linked with healthcare settings, owing to its unique ability to survive killing by disinfectants and antimicrobials and to withstand desiccation on dry surfaces (11). Intensive care units (ICU) in resource-limited settings can be epicenters of environmental CRAb proliferation due to limited infection prevention and control (IPC) capacity to detect reservoirs and interrupt transmission among patients who may be immunocompromised and have indwelling medical devices (12). Molecular analyses have been used to detect protracted outbreaks of CRAb in settings where it has become hyperendemic and have helped to shed light on likely reservoirs and transmission pathways (13–16). However, few LMIC settings have utilized temporal sequencing of clinical and environmental CRAb isolates to identify outbreaks and transmission pathways. Whole genome sequencing (WGS) is becoming increasingly accessible in LMICs and represents a promising tool for outbreak detection in LMICs when traditional epidemiologic approaches fall short.

In this study, we conducted a WGS analysis on clinical and environmental *A. baumannii* isolates collected from a tertiary referral hospital in Botswana where there was evidence of increasing rates of CRAb infections among neonates. We also identified diagnostic genes that can be used, without the need for WGS, to determine whether future CRAb isolates from this hospital are members of the outbreak clones.

## MATERIALS AND METHODS

### Study design and setting

We performed WGS on all viable, stored, presumptive *A. baumannii* isolates collected from 1 March 2021 to 31 August 2022 at a 530-bed public tertiary referral hospital in Botswana with an 8-bed ICU and a 33-bed neonatal intensive care unit (NICU). During this period, *A. baumannii* isolate storage was coordinated by a laboratory-based surveillance program which had undergone ethical review by Institutional Review Boards (IRBs) at the University of Pennsylvania (IRB# 2022-851492), the University of Botswana, the Health Research and Development Committee at Botswana's Ministry of Health (2022-HPDME18/13/1), and the study healthcare facility (2022-2/2A (7)/201). De-identified isolates were submitted to the National Institute for Communicable Diseases (NICD; Johannesburg, South Africa) for WGS. We obtained a waiver of written patient consent from the IRBs due to this being a retrospective abstraction of surveillance data.

## Clinical isolates

Blood cultures were collected by hospital staff following appropriate skin antisepsis from patients when clinically indicated due to a suspicion of sepsis. Samples were then incubated at the hospital microbiology laboratory using an automated system (BACT/ALERT, BioMérieux). Isolate identification was performed manually using Gram stain and colony morphology, followed by sub-culturing and analytical profile index strip (BioMérieux) and biochemical testing. Antimicrobial susceptibility testing using the disc diffusion method at the hospital microbiology laboratory was done inconsistently during this period given the frequent stock-outs of antibiotic discs. Blood cultures were classified as healthcare-associated if they were collected after more than 72 h of hospitalization or, in the case of inborn, continuously hospitalized neonates, after the first day of life (17). All available stored blood culture isolates previously identified as *A. baumannii* were sub-cultured prior to submission for sequencing.

## NICU environmental isolates

Environmental isolates were collected during a series of four point prevalence surveys conducted over 6 months (January through June 2021) which aimed to identify reservoirs of multidrug-resistant organisms in the NICU environment (18). Sites sampled included high-touch surfaces, sink drains and water, equipment, and hands of caregivers and healthcare workers. Samples were collected using sterile flocked swabs, and sampling sites were chosen without regard to epidemiologic links to patients with infection. Samples were transported in sterile water and individually inoculated onto chromogenic media (CHROMagar ESBL, Paris, France) within 24 h for the selection and differentiation of extended-spectrum β-lactamase-producing organisms. Colonies were identified using visual inspection of chromogenic media only according to the manufacturer's guidelines. Isolates presumptively identified as *Acinetobacter* spp. from banked environmental specimens were sub-cultured and submitted for sequencing.

## Whole genome sequencing

Following initial phenotypic identification by the hospital's clinical microbiology laboratories, all clinical and environmental isolates were stored at −80°C until shipment of isolates to NICD laboratories. At NICD, genomic DNA was extracted from bacteria using an Invitrogen PureLink Microbiome DNA Purification Kit (Invitrogen, USA). To assess environmental and reagent contamination, we included a reagent-only negative control processed alongside the samples, which underwent the same extraction procedure but contained no bacterial culture. DNA concentration in the negative control, measured using a NanoDrop spectrophotometer (Thermo Scientific), was 2.63 ng/μL, below the expected threshold of 10 ng/μL. Shotgun libraries were generated using the Illumina DNA Prep Library Prep kit (Illumina). Successful libraries were sequenced on an Illumina NextSeq, producing $2 \times 150$ base pair paired-end reads with ~80 times coverage (File S1). All WGS methodologies were performed using the protocols and standard operating procedures as prescribed by the manufacturers of the kits and equipment. All sequencing data generated in this study were deposited on 29 February 2024 in NCBI under BioProject PRJNA1082310.

## Genome and phylogenetic analysis

Taxonomic assignment (19), MLST using the Pasteur scheme (20), antimicrobial resistance gene identification (21), and phylogenetic analyses were performed using publicly accessible analysis pipelines at NICD and at the University of Pennsylvania/Children's Hospital of Philadelphia Microbiome Center. A WhatsGNU database for *A. baumannii* was constructed from 14,627 publicly available genomes downloaded using National Center for Biotechnology Information genome download (22). The genomes were processed using the bactopia pipeline v2.2.0 (23), and *de novo* assembly was completed using Shovill v1.1.0 (24). Sequence type of the genomes was determined using mlst

v2.19.0 (20), which made use of the PubMLST website (25). K and O locus typing was performed using Kaptive v. 2.0.1 with default settings applied to assembled genomes, yielding locus assignments with associated identity, coverage, and confidence metrics (26, 27). A maximum likelihood tree was built using the Cladebreaker pipeline (28) and included 28 genomes from our collection and 16 assembled genomes available on GenBank (29), chosen using the topgenome (-t) feature of WhatsGNU with three top genomes specified (30). Within Cladebreaker, genome annotation was completed using Prokka v1.14.6 (31), and a core genome nucleotide alignment produced by Roary v3.13.0 (32). This alignment was then used to infer an initial phylogenetic tree in RAxML v8.2.9 using GTR substitution model accounting for among-site rate heterogeneity using the Γ distribution and four rate categories (GTRGAMMA model) for 100 individual searches with maximum parsimony random-addition starting trees (33–35). Node support was evaluated with 100 nonparametric bootstrap pseudoreplicates (36). A second maximum likelihood tree was created for the sequence type 1 (ST1) genomes using a Snippy SNP alignment to a reference that was the genome of best quality genome from our collection (ST1-FMW-BSI-Jun-22) (37). The SNP differences from both phylogenetic matrices were determined using SNP-dists (38). There are no universal thresholds for determining whether any two CRAb strains are part of the same outbreak, but SNP differences of less than three have been used to conservatively define isolates as unequivocally part of an intrahospital outbreak (39). The data used in this publication were collected through the MENDEL high performance computing cluster at the American Museum of Natural History.

## Diagnostic genes and alleles

Roary and Scoary were used to identify genes uniquely present or absent in each clade (32, 40). We used annotated assemblies in GFF3 format produced by Prokka to calculate the pan-genome in Roary (31). The output from Roary was used by Scoary to identify statistically significant genes found in each clade compared to the other clades. The DAMAGE application, which integrates WhatsGNU output, was used to identify gene protein alleles specific to each clade (41). This was used to sort alleles according to clade and execute statistical tests (sensitivity, specificity, p-value, odds-ratio, Bonferroni correction) similar to Roary. The WhatsGNU database composed of 14,627 publicly available genomes was used to identify genomes in publicly available databases with exact matches to the diagnostic alleles (-i, --ids_hits option). We required each allele to have 100% specificity and 100% sensitivity for the genomes in each clade, and a gene novelty unit score of 0 indicating that this exactly matched allele is never seen in the *A. baumannii* whole genome database (*n* = 14,627 genomes).

## RESULTS

### Characterization of isolates

Of the 54 blood cultures reporting growth of *A. baumannii* during the surveillance period, a total of 23 isolates were available for sequencing (remaining isolates had been discarded). Cultures had been taken from patients aged 2 days to 69 years who were admitted in the ICU (*n* = 6), NICU (8), Male Medical Ward (*n* = 3), Female Medical Ward (*n* = 2), and Emergency Department (*n* = 4). Most cases (82%; 19/23) were healthcare-associated, the remaining four were community-associated. Sequencing quality for *Acinetobacter* spp. was generally high with average coverage of 165x (range 89x–235x), and an average N50 of 149,428 (range 49,974–566,324) (File S1). Assemblies ranged from 3,731,429 to 4,023,588 base pairs. All 23 blood culture isolates were confirmed to be *A. baumannii* by WGS. Out of 20 environmental isolates presumptively identified as *A. baumannii* using chromogenic media, only 25% (*n* = 5) were confirmed as *A. baumannii*. The remaining environmental isolates were identified as other *Acinetobacter* spp. (*n* = 3), *Serratia* spp. (*n* = 6), *Pseudomonas* spp. (*n* = 3), *Exiguobacterium* spp. (*n* = 1), *Microbacterium* spp. (*n* = 1), and one unclassified organism.

| Antimicrobial / Biocide Class | | β-lactams | | | | | | | | | | | Aminoglycosides | | | | | | Macrolides | | Sulfonamides | | Tetracyclines | Trimethoprim | Biocide | Phenicol | Rifampicin |
|---|---|---|---|---|---|---|---|---|---|---|---|---|---|---|---|---|---|---|---|---|---|---|---|---|---|---|---|
| **Strain Name** | **Sequence Type** | blaADC-25 | blaNDM-1 | blaOXA-23 | blaOXA-64 | blaOXA-65 | blaOXA-69 | blaOXA-88 | blaOXA-120 | blaOXA-203 | blaOXA-343 | blaOXA-314 | aadA1 | aac(3)-Ia | armA | aph(3')Ia | aph(3'')-Ib | aph(6)-Id | msr(E) | mph(E) | sul1 | sul2 | tet(B) | dfrA1 | qacE | cmlA1 | ARR-2 |
| ST1-NICU-BSI-Apr-21 | 1 | ✓ | ✓ | ✓ | | | ✓ | | | | | | ✓ | ✓ | ✓ | ✓ | ✓ | ✓ | ✓ | ✓ | ✓ | ✓ | ✓ | ✓ | ✓ | ✓ | ✓ |
| ST1-ICU-BSI-Aug-22-2 | 1 | ✓ | ✓ | ✓ | | | ✓ | | | | | | ✓ | ✓ | ✓ | | | | ✓ | ✓ | ✓ | ✓ | ✓ | ✓ | ✓ | ✓ | ✓ |
| ST1-ICU-BSI-Aug-22-1 | 1 | ✓ | ✓ | ✓ | | | ✓ | | | | | | ✓ | ✓ | ✓ | | | ✓ | ✓ | ✓ | ✓ | ✓ | | ✓ | ✓ | ✓ | ✓ |
| ST1-MMW-BSI-Aug-22-2 | 1 | ✓ | ✓ | ✓ | | | ✓ | | | | | | ✓ | ✓ | ✓ | ✓ | ✓ | ✓ | ✓ | ✓ | ✓ | ✓ | ✓ | ✓ | ✓ | ✓ | ✓ |
| ST1-MMW-BSI-Aug-22-1 | 1 | ✓ | ✓ | ✓ | | | ✓ | | | | | | ✓ | ✓ | ✓ | ✓ | ✓ | ✓ | ✓ | ✓ | ✓ | ✓ | ✓ | ✓ | ✓ | ✓ | ✓ |
| ST1-NICU-Env-Apr-21-2 | 1 | ✓ | ✓ | ✓ | | | ✓ | | | | | | ✓ | ✓ | ✓ | ✓ | ✓ | ✓ | ✓ | ✓ | ✓ | ✓ | ✓ | ✓ | ✓ | ✓ | ✓ |
| ST1-NICU-Env-Apr-21-1 | 1 | ✓ | ✓ | ✓ | | | ✓ | | | | | | ✓ | ✓ | ✓ | ✓ | ✓ | ✓ | ✓ | ✓ | ✓ | ✓ | ✓ | ✓ | ✓ | ✓ | ✓ |
| ST1-NICU-Env-Apr-21-3 | 1 | ✓ | ✓ | ✓ | | | ✓ | | | | | | ✓ | ✓ | ✓ | ✓ | ✓ | ✓ | ✓ | ✓ | ✓ | ✓ | ✓ | ✓ | ✓ | ✓ | ✓ |
| ST1-NICU-BSI-Mar-22 | 1 | ✓ | ✓ | ✓ | | | ✓ | | | | | | ✓ | ✓ | ✓ | ✓ | ✓ | ✓ | ✓ | ✓ | ✓ | ✓ | ✓ | ✓ | ✓ | ✓ | ✓ |
| ST1-FMW-BSI-Mar-21 | 1 | ✓ | ✓ | ✓ | | | ✓ | | | | | | ✓ | ✓ | ✓ | | ✓ | ✓ | ✓ | ✓ | ✓ | ✓ | ✓ | ✓ | ✓ | ✓ | ✓ |
| ST1-ICU-BSI-Mar-21 | 1 | ✓ | ✓ | ✓ | | | ✓ | | | | | | ✓ | ✓ | | | ✓ | ✓ | | | ✓ | ✓ | ✓ | ✓ | | | |
| ST1-AE-BSI-Jun-21 | 1 | ✓ | ✓ | ✓ | | | ✓ | | | | | | ✓ | ✓ | ✓ | ✓ | ✓ | ✓ | ✓ | ✓ | ✓ | ✓ | ✓ | ✓ | ✓ | ✓ | ✓ |
| ST1-ICU-BSI-Jun-22 | 1 | ✓ | ✓ | ✓ | | | ✓ | | | | | | ✓ | ✓ | ✓ | | ✓ | ✓ | ✓ | ✓ | ✓ | ✓ | ✓ | ✓ | ✓ | ✓ | ✓ |
| ST1-FMW-BSI-Jun-22 | 1 | ✓ | ✓ | ✓ | | | ✓ | | | | | | ✓ | ✓ | ✓ | ✓ | ✓ | ✓ | ✓ | ✓ | ✓ | ✓ | ✓ | ✓ | ✓ | ✓ | ✓ |
| ST1-NICU-BSI-Jul-21 | 1 | ✓ | ✓ | ✓ | | | ✓ | | | | | | ✓ | ✓ | ✓ | ✓ | ✓ | ✓ | ✓ | ✓ | ✓ | ✓ | ✓ | ✓ | ✓ | ✓ | ✓ |
| ST1-MMW-BSI-Aug-22 | 1 | ✓ | ✓ | ✓ | | | ✓ | | | | | | ✓ | ✓ | ✓ | ✓ | ✓ | ✓ | ✓ | ✓ | ✓ | ✓ | ✓ | ✓ | ✓ | ✓ | ✓ |
| ST1-ICU-BSI-Aug-22 | 1 | ✓ | ✓ | ✓ | | | ✓ | | | | | | ✓ | ✓ | ✓ | | ✓ | ✓ | ✓ | ✓ | ✓ | ✓ | ✓ | ✓ | ✓ | ✓ | ✓ |
| ST1-NICU-BSI-Oct-21-1 | 1 | ✓ | ✓ | ✓ | | | ✓ | | | | | | ✓ | ✓ | ✓ | ✓ | ✓ | ✓ | ✓ | ✓ | ✓ | ✓ | ✓ | ✓ | ✓ | ✓ | ✓ |
| ST1-NICU-BSI-Nov-21-1 | 1 | ✓ | ✓ | ✓ | | | ✓ | | | | | | ✓ | ✓ | ✓ | ✓ | ✓ | ✓ | ✓ | ✓ | ✓ | ✓ | ✓ | ✓ | ✓ | ✓ | ✓ |
| ST1-NICU-BSI-Nov-21-2 | 1 | ✓ | ✓ | ✓ | | | ✓ | | | | | | ✓ | ✓ | ✓ | ✓ | ✓ | ✓ | ✓ | ✓ | ✓ | ✓ | ✓ | ✓ | ✓ | ✓ | ✓ |
| ST1-NICU-BSI-Dec-21-2 | 1 | ✓ | ✓ | ✓ | | | ✓ | | | | | | ✓ | ✓ | ✓ | ✓ | ✓ | ✓ | ✓ | ✓ | ✓ | ✓ | ✓ | ✓ | ✓ | ✓ | ✓ |
| ST1-NICU-BSI-Dec-21-1 | 1 | ✓ | ✓ | ✓ | | | ✓ | | | | | | ✓ | ✓ | ✓ | ✓ | ✓ | ✓ | ✓ | ✓ | ✓ | ✓ | ✓ | ✓ | ✓ | ✓ | ✓ |
| ST152-NICU-Env-Apr-21-2 | 152 | ✓ | | ✓ | | | | | | | | | | | | | | | | | | | | | | | | |
| ST152-NICU-Env-Apr-21-1 | 152 | ✓ | | ✓ | | | | | | | | | | | | | | | | | | | | | | | | |
| ST193-AE-BSI-Apr-21 | 193 | ✓ | | | | | ✓ | | | | | | | | | | | | | | | | | | | | | |
| ST976-AE-BSI-Aug-22 | 976 | ✓ | | | | | | ✓ | | | | | | | | | | | | | | | | | | | | |
| ST1438-AE-BSI-May-22 | 1438 | ✓ | | | ✓ | ✓ | | | ✓ | | | | | | | | | | | | | | | | | | | |
| UI-BSI-ICU-May-22 | Unidentified | ✓ | | | | | | | | ✓ | | | | | | | | | | | | | | | | | | |

**FIG 1** Antimicrobial/biocide resistance genes identified in clinical and environmental *Acinetobacter baumannii* isolates collected from March 2021–August 2022.

## Sequence typing, capsular polysaccharide locus typing, and AMR gene characterization

MLST analysis revealed that 79% (22/28) of *A. baumannii* isolates were ST1, including all 19 healthcare-associated blood isolates and 60% (3/5) of environmental isolates. Phylogenetic analysis confirmed the monophyly of these 22 genomes (File S1). Kaptive identified six distinct K loci, with KL17 predominating (22/28, 79%) (File S7). O-locus typing revealed four variants, with OCL1 detected in 24/28 isolates (86%), most commonly in combination with KL17 (22/28, 79%). Minority pairings included KL32:OCL21 (*n* = 2), KL46:OCL4 (*n* = 1), KL142:OCL6 (*n* = 1), KL79:OCL1 (*n* = 1), and KL155:OCL1 (*n* = 1).

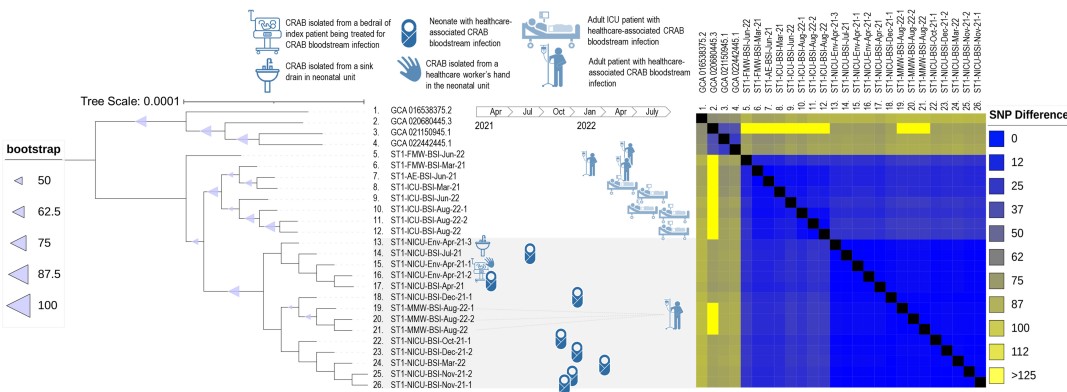

**FIG 2** Maximum Likelihood Phylogenetic tree (left) with nodes labeled with bootstrap values (size of triangle is proportional to value) and pairwise single nucleotide polymorphism differences (right) of sequence type 1 (ST1) clinical and environmental carbapenem-resistant *Acinetobacter baumannii* (CRAb) isolates collected from March 2021–August 2022. Scale bar is in substitutions/site.

ST1, which represented the dominant lineage in our collection, carried a broad suite of resistance determinants including multiple β-lactamases (*bla*) (*bla*$_{OXA-69}$, *bla*$_{OXA-23}$, *bla*$_{NDM-1}$, and *bla*$_{ADC-25}$, of which *bla*$_{OXA-23}$, *bla*$_{NDM-1}$ are genes encoding carbapenemases), as well as genes conferring resistance to aminoglycosides, tetracyclines, sulfonamides, trimethoprim, and macrolides (Fig. 1). This constellation of ARGs is consistent with extensively drug-resistant (XDR) phenotypes described for global clone 1 (GC1) strains. Genes encoding biocide resistance (quaternary ammonium compound E [*qacE*]) were present in 100% (*n* = 22) of ST1 isolates.

None of the four community-associated blood isolates were identified as ST1, and none were related to each other (ST976 containing *bla*$_{OXA-203}$, ST193 containing *bla*$_{OXA-120}$, ST1438 containing *bla*$_{OXA-88}$, *bla*$_{OXA-65}$, *bla*$_{OXA-343}$, and one unidentified MLST containing *bla*$_{OXA-314}$). No plasmid-mediated colistin resistance genes (mcr family) were detected. The other two environmental isolates were identified as ST152 and contained the carbapenemase encoding gene, *bla*$_{OXA-64}$.

## Phylogenetic analysis

Phylogenetic analysis of the ST1 clade demonstrated spatial clustering by hospital unit. Very closely related isolates spanned wide ranges in time (>1 year), suggesting ongoing transmission from environmental sources (Fig. 2; File S1) (42). We computed SNP distance using two datasets, a reference-based data set produced by Snippy (Fig. 2; File S2) and a core gene nucleotide data set produced by Roary and Scoary (File S3) (32, 37, 40). The reference-based data set showed a clade (average difference 2.3, range 0–5, SNPs) containing all eight neonatal blood isolates and also the three environmental isolates from the neonatal unit (a sink drain, bed rail, and a healthcare worker's hand swab). Three isolates from one adult patient (obtained from three separate blood cultures sent on the same day) were also included in this clade; however, there was no clear epidemiologic link identified among the neonatal patients. The tree also suggested the existence of a local clade that contains both the neonatal clade and other isolates from the same hospital (average difference 13.8, range 0–30 SNPs). The core genome data set also produced very similar SNP differences (neonatal clade average SNPs = 0.16, local clade SNPs = 4.9). Reference-based ST1 genomes were accessed at https://itol.embl.de/tree/7116771201067167336999 and the pangenome (all STs) at https://itol.embl.de/tree/711677120439081671553811.

## Diagnostic alleles

The neonatal clade was characterized by 11 such protein alleles with various predicted functions, and the local ST1 clade that included the neonatal clade was characterized by

10 protein alleles most of which had no predicted function (Files S4 and S5). Nucleotide and amino acid sequences for each of these genes is included as supplementary data (File S6).

## DISCUSSION

We detected a protracted hospital CRAb outbreak with transmission spanning at least 18 months using WGS. All healthcare-associated strains shared the same clonal lineage and many had been isolated from newborns who had never left the hospital, suggesting that ongoing transmission was facilitated by the presence of reservoirs in the hospital environment. Although only five environmental *A. baumannii* isolates were available for analysis, three shared the same clonal lineage as the clinical isolates. These data, in combination with the spatiotemporal association between the environmental isolates and clinical cases, suggest that sink drains were likely reservoirs, while surfaces and hands of healthcare workers may have served as transmission vehicles (43).

Although CRAb-contamination of hospital sinks has been reported elsewhere in Africa (44), our study is among the first in an LMIC setting to utilize temporally linked sequencing of clinical and environmental isolates to demonstrate that sink drains are a likely reservoir for a sustained clinical transmission (43). During a CRAb outbreak at a tertiary hospital in Lebanon, WGS was used to confirm that the CRAb outbreak ($n = 41$) was polyclonal in nature but did not include environmental samples as part of the investigation (14). An investigation of *A. baumannii* cases from a Latvian neonatal unit spanning a period of five years ($n = 17$) which incorporated environmental sampling did not find environmental *A. baumannii* contamination and used WGS to determine polyclonal origins of isolates (16).

These are the first *A. baumannii* genomes reported originating from Botswana. We identified the carbapenemases $bla_{OXA-23}$ and $bla_{NDM-1}$ in all 19 ST1 isolates. $bla_{OXA-23}$ has been well-described as one of the most common carbapenemases identified among CRAb isolates globally due, in part, to its location on mobile genetic elements (45, 46); according to a systematic review of CRAb reports from sub-Saharan Africa, with most studies originating from South Africa, the most frequently identified carbapenemase genes are $bla_{OXA-23}$ and $bla_{VIM}$ (47). However, its co-occurrence with $bla_{NDM-1}$ is a more recent phenomenon. $bla_{NDM-1}$ likely originated in *A. baumannii* (48) and was first recognized in patients from south Asia, but in recent years, it has been identified in clinical CRAb isolates from north, east, west, and southern Africa (49, 50). The constellation of $bla_{OXA-23}$ and $bla_{NDM-1}$ together with aminoglycoside methyltransferase armA and Tn7-family insertions is consistent with features described for the MRSN56 sublineage and its evolved KL17/Aci-IE1 variant. Although we did not identify the *mcr*-1 gene known to confer colistin-resistance and phenotypic colistin sensitivity testing was not performed (due to local unavailability), a 2022 South African study found that the majority of colistin-resistant CRAb isolates ($n = 6$) were ST1 harboring both $bla_{OXA-23}$ and $bla_{NDM-1}$, and notably, these isolates also carried the KL17:OCL1 locus combination—consistent with the dominant clone identified in our study (51).

Although the isolates sequenced generally clustered spatially by ward, their shared clonal lineage (ST1) and the case of an adult patient closely related to the "neonatal" clade raises questions around shared exposures across wards and reservoirs yet to be identified. These cross-unit exposures could include healthcare workers working in both wards, shared cleaning equipment (e.g., mops and buckets), or shared water and plumbing systems. Of note, no *A. baumannii* was isolated from fresh tap water samples obtained from water taps serving the NICU sinks during the environmental point prevalence surveys from which this analysis drew its environmental isolates although the methods used in these surveys lacked the level of sensitivity needed to definitely rule out tap water as a potential source (18).

Sinks, although important for hand hygiene, are often used for other purposes such as utensil cleaning and for food and medicine disposal and can harbor robust biofilms and serve as a reservoir for multidrug-resistant organisms (52–54). These difficult-to-dislodge

biofilms necessitate reservoir remediation strategies, especially for hospitals in resource-limited settings. Some evidence supports radical measures to eliminate water-related reservoirs including "water-free patient care" (i.e., complete removal of sinks from patient care areas) (55). However, these interventions have not been well-tested in high-resource settings, and their feasibility in resource-limited settings is unclear. Alternative approaches, such as sink re-design (e.g., plumbing trap stop valves, boiling water, splashguards) and remediation strategies (e.g., thermal, chemical, or phage-based), may be more easily implemented in these settings (56–58).

The lack of strain diversity over an extended period may suggest that *Acinetobacter* spp. are capable of persisting in hospital environmental reservoirs despite competition from other organisms or use of disinfectants. Aside from reservoir remediation strategies, there is a general need for rapid adaptation of IPC protocols appropriate for resource-limited settings. Use of invasive medical devices in LMICs has outpaced the IPC measures needed to maintain and safely reprocess them and the global healthcare worker shortage has resulted in too few staff to implement IPC measures (59). Multimodal IPC bundles are needed which deploy a combination of optimization of hand hygiene, colonization-based patient screening, environmental sampling, and enhanced terminal and non-terminal cleaning and disinfection (60).

This analysis highlights the power of WGS to shed light on complex public health problems in LMICs bearing the highest burden of AMR. Modeling studies in high-income countries estimate that the use of WGS early on in CRAb outbreak investigations results in significant cost savings, fewer infections, and additional quality-adjusted life years (46). As WGS technologies become less expensive and open-source analytic tools more accessible, broad support for the adaptation of WGS in public health laboratories in LMICs is needed (61). South-South and North-South collaborations are pivotal in supporting technical development and in ensuring sustainable implementation of WGS into national action plans to combat AMR.

Even as LMICs adopt WGS technology into the public health laboratory ecosystem, alternative, low-cost strategies are needed for molecular tracking in the midst of outbreaks. Identification of alleles that are highly specific and sensitive markers for specific clades is an emerging field promising a low-cost method of tracking outbreak strains and identifying potential reservoirs (62–64). There are two major benefits to this diagnostic approach. First, new isolates can be screened for inclusion in each previously identified clade without full-scale phylogenetic analysis. Second, amplification of these genes and sequencing by standard Sanger sequencing can be done at a fraction of the cost of WGS. Even more efficient, polymerase chain reaction primers could be designed to amplify these genes directly from clinical or environmental samples without the need for culture. Altogether, the identification of diagnostic alleles in this outbreak investigation serves as a proof of principle that a limited amount of WGS can be used to develop less expensive and more rapidly deployable molecular tools for surveillance and epidemiology.

## Limitations

This study was limited by the small number of clinical samples (23/54 of blood isolates) and an even smaller number of environmental *A. baumannii* isolates available for sequencing. Identifying genetically linked isolates from patient and environmental samples does not prove the sampled site is the definitive point-source. Because this clonal type was recovered from multiple environmental sites, it is likely that there may have been additional reservoirs and vehicles not identified because of the limited scope of the environmental sampling exercise, thus obscuring the full spectrum of transmission. Future investigations would incorporate more samples and would prioritize sites epidemiologically linked to patients with confirmed infection. Additionally, the use of chromogenic media to screen for *A. baumannii* in environmental samples had low specificity; future studies would take advantage of media designed specifically for *A.*

*baumannii* recovery, as well as automated identification of environmental isolates prior to sequencing.

## Conclusions

This report demonstrates the power of WGS to define true outbreaks in the setting of hyperendemic spread, a scenario which is common in LMIC and when traditional epidemiologic approaches fall short. Additionally, our work makes a case for public health and reference laboratories in LMICs to be equipped with sequencing technologies so WGS can be leveraged in the regions bearing the highest burden of AMR and struggle to understand the drivers of hyperendemic and outbreak spread of these organisms. South-South and North-South partnerships like the ones described in this manuscript are critical in building public health laboratory infrastructure to detect and contain health threats caused by AMR. This work also highlights the importance of environmental sampling to identify and remediate reservoirs (e.g., sink drains) and vehicles (e.g., hands, surfaces) within the healthcare environment and is a call-to-action to develop remediation strategies and provide broader support for IPC teams in resource-limited settings.

## ACKNOWLEDGMENTS

Whole genome sequencing of bacterial isolates was made possible by the SEQAFRICA project supported by the Department of Health and Social Care's Fleming Fund using UK aid. The views expressed in this publication are those of the authors and not necessarily those of the UK Department of Health and Social Care or its Management Agent, Mott MacDonald. The high performance computing cluster described in this report was developed with National Science Foundation Campus Cyberinfrastructure support through Award#1925590.

## AUTHOR AFFILIATIONS

[1]Department of Paediatric & Adolescent Health, Faculties of Medicine & Health Sciences, University of Botswana, Gaborone, Botswana

[2]Children's Hospital of Philadelphia, Philadelphia, Pennsylvania, USA

[3]Botswana-University of Pennsylvania Partnership, Gaborone, Botswana

[4]Perelman School of Medicine, University of Pennsylvania, Philadelphia, Pennsylvania, USA

[5]Ministry of Health and Wellness, Gaborone, Botswana

[6]Department of School of Allied Health Professions, Faculties of Medicine & Health Sciences, University of Botswana, Gaborone, Botswana

[7]Biomedical Sciences, Faculties of Medicine & Health Sciences, University of Botswana, Gaborone, Botswana

[8]Department of Pathology and Laboratory Medicine, University of British Columbia, Vancouver, British Columbia, Canada

[9]The Hospital for Sick Children, Toronto, Ontario, Canada

[10]Department of Laboratory Medicine and Pathobiology, University of Toronto, Toronto, Canada

[11]National Institute for Communicable Diseases, Johannesburg, South Africa

[12]Department of Medical Microbiology, Faculty of Health Sciences, University of Pretoria, Pretoria, South Africa

## AUTHOR ORCIDs

Jonathan Strysko  http://orcid.org/0000-0002-5258-7654
Kwana Lechiile  http://orcid.org/0000-0001-6691-1905
David M. Goldfarb  http://orcid.org/0000-0003-0835-9504
Melissa Richard-Greenblatt  http://orcid.org/0000-0001-6717-8856

Paul J. Planet 🔘 http://orcid.org/0000-0003-0543-0539

## AUTHOR CONTRIBUTIONS

Jonathan Strysko, Conceptualization, Data curation, Formal analysis, Funding acquisition, Investigation, Methodology, Project administration, Resources, Supervision, Visualization, Writing – original draft, Writing – review and editing | Tefelo Thela, Data curation, Methodology, Project administration, Writing – review and editing | Andries Feder, Formal analysis, Software, Visualization, Writing – review and editing | Janet Thubuka, Data curation, Investigation, Methodology, Writing – review and editing | Tichaona Machiya, Data curation, Investigation, Methodology, Writing – review and editing | Jack Mkubwa, Investigation, Writing – review and editing | Kagiso Mochankana, Investigation, Writing – review and editing | Celda Tiroyakgosi, Methodology, Writing – review and editing | Kgomotso Kgomanyane, Investigation, Methodology, Writing – review and editing | Tshiamo Zankere, Data curation, Investigation, Writing – review and editing | Kwana Lechiile, Investigation, Writing – review and editing | Teresia Gatonye, Investigation, Writing – review and editing | Chimwemwe Viola Tembo, Investigation, Writing – review and editing | Moses Vurayai, Investigation, Methodology, Writing – review and editing | Naledi Mannathoko, Conceptualization, Supervision, Writing – review and editing | Margaret Mokomane, Conceptualization, Supervision, Writing – review and editing | Ahmed M. Moustafa, Formal analysis, Methodology, Software, Writing – review and editing | David M. Goldfarb, Conceptualization, Methodology, Writing – review and editing | Melissa Richard-Greenblatt, Conceptualization, Methodology, Supervision, Writing – review and editing | Carolyn McGann, Methodology, Writing – review and editing | Susan E. Coffin, Conceptualization, Formal analysis, Methodology, Supervision, Writing – original draft, Writing – review and editing | Britt Nakstad, Writing – review and editing | Corrado Cancedda, Writing – review and editing | Dineo Bogoshi, Data curation, Formal analysis, Investigation, Methodology, Software, Writing – review and editing | Anthony M. Smith, Data curation, Formal analysis, Funding acquisition, Methodology, Software, Writing – review and editing | Paul J. Planet, Conceptualization, Data curation, Formal analysis, Funding acquisition, Methodology, Software, Supervision, Visualization, Writing – review and editing.

## ADDITIONAL FILES

The following material is available online.

### Supplemental Material

**File S1 (Spectrum01768-25-s0001.xlsx).** Sequence run identifiers, sequence types, and resistance gene hits.
**File S2 (Spectrum01768-25-s0002.xlsx).** Pairwise SNP distance data and summary statistics.
**File S3 (Spectrum01768-25-s0003.xlsx).** SNP distance matrix.
**File S4 (Spectrum01768-25-s0004.csv).** Allele-level output from the DAMAGE pipeline showing diagnostic genes specific to the neonatal clade of *A. baumannii*.
**File S5 (Spectrum01768-25-s0005.xlsx).** Diagnostic genes and alleles specific to the ST1 clade.
**File S6 (Spectrum01768-25-s0006.txt).** FASTA-format nucleotide sequences representing unique outbreak-specific alleles identified through the DAMAGE pipeline, supporting molecular tracking of *A. baumannii* outbreak strains
**File S7 (Spectrum01768-25-s0007.xlsx).** K- and O- loci of *Acinetobacter baumannii* isolates (KAPTIVE).

## Open Peer Review

**PEER REVIEW HISTORY (review-history.pdf).** An accounting of the reviewer comments and feedback.

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
