## [Reviewer comments · Microbiology Spectrum]

Microbiology Spectrum

Carbapenem-resistant *Acinetobacter baumannii* at a hospital in Botswana: Detecting a protracted outbreak using whole genome sequencing

Jonathan Stryko, Tefelo Thela, Andries Feder, Janet Thubuka, Tichaona Machiya, Jack Mkubwa, Kagiso Mochankana, Celda Tiroyakgosi, Kgomotso Kgomonyane, Tlhalefo Ntereke, Tshiamo Zankere, Kwana Lechiile, Teresia Gatonye, Chimwemwe Tembo, Moses Vurayai, Naledi Mannathoko, Margaret Mokomane, Ahmed Moustafa, David Goldfarb, Melissa Richard-Greenblatt, Carolyn McGann, Susan Coffin, Britt Nakstad, Corrado Cancedda, Ebbing Lautenbach, Dineo Bogoshi, Anthony Smith, and Paul Planet

Corresponding Author(s): Jonathan Stryko, Botswana-University of Pennsylvania Partnership

Review Timeline:

Submission Date:	June 10, 2025
Editorial Decision:	August 7, 2025
Revision Received:	September 10, 2025
Editorial Decision:	October 16, 2025
Revision Received:	October 17, 2025
Accepted:	October 22, 2025

Editor: Ruth Hall

Reviewer(s): The reviewers have opted to remain anonymous.

Transaction Report:

DOI: <https://doi.org/10.1128/spectrum.01768-25>

Re: Spectrum01768-25 (Carbapenem-resistant *Acinetobacter baumannii* at a hospital in Botswana: Detecting a protracted outbreak using whole genome sequencing)

Dear Dr. Jonathan Stryko:

Thank you for the privilege of reviewing your work. Below you will find my comments, instructions from the Spectrum editorial office, and the reviewer comments.

Please follow the directions of the reviewer who has made valuable suggestions. In particular move information into the main text where possible and include a data availability statement about the accession numbers for genomes so that readers can find them.

Revision Guidelines

Sincerely,
Ruth Hall
Editor
Microbiology Spectrum

Reviewer #1 (Comments for the Author):

The manuscript by Stryko et. al. describes an analysis of carbapenem-resistant *A. baumannii* in a Botswanan hospital setting. This is an important contribution to the field, as it represents the first *A. baumannii* genomes to originate from Botswana.

The manuscript is generally well written and includes a nicely executed genomic examination of the isolates recovered within the hospital setting. However, there are several improvements which would strengthen the manuscript and contextualise the collection of isolates against the broader *A. baumannii* global story.

1) The authors may be interested in this manuscript (DOI 10.1038/s44259-025-00103-5) by Harmer and colleagues. At a glance, it looks like the authors strains may be related to the lineage reported by Harmer et. al. The combination of resistance genes in these Botswanan isolates is consistent with the isolates having the *Aci-IE1* and *Tn7++* islands reported by Harmer. Determining the KL (see point 3 below) would add further evidence to this.

2) A strength of this manuscript is that these are the first *A. baumannii* genomes from Botswana, so the authors should highlight that. It isn't clear from the manuscript text whether these genomes have been released in GenBank, this information can only be gleaned from examining the supplementary table. Please include this information in the main text, possibly as a table of accession numbers.

3) Please determine (and report) the capsule (KL) and outer (OCL) types for these isolates. This is readily achievable using Kaptive and would be highly informative when comparing against other ST1 outbreaks.

4) There is a lot of valuable information buried in the supplementary file that should be included in the main text to further strengthen the manuscript. For example, the main text only makes mention of the *bla*NDM and *oxa23* resistance genes, but these isolates include a suite of additional resistance genes including the clinically important *armA* aminoglycoside resistance gene. Presenting the full resistance gene arsenal for each isolate in a table (or presented visually) in the main text would significantly strengthen the manuscript.

5) The "Characterisation of isolates" section is quite confusing and would benefit significantly from a table which characterises the key features of each isolate. For example, this section refers to an assembly of 3,031,626 bp which seems alarmingly small for an *A. baumannii*. Reading further, and correlating against the supplementary table, it becomes clearer that this is one of the non-*baumannii* environmental isolates.

Minor points

1) Line 147. Please report the date at which the genome download was performed.

2) Line 206. The statement "Mobile colistin resistance-1 gene mutations were not identified" is confusing and unnecessary. The isolates do not have *mcr-1* (or any other colistin resistance gene from what I can tell), so they would not have gene mutations.

3) Figure 1. What are the four reference genomes included in the phylogeny? These need to be identified.

Comments to the authors:

Methods section:

AST inconsistency due to stock-outs is an important limitation. Was any AST data available at all?

Please state the rationale for choosing the study period (March 2021 to August 2022). Was it based on an outbreak, surveillance funding, or other factors?

The API for non-fermenters used in the identification is not enough, automated machine as VITEK was not employed for identification??

I believe that the use of CHROMagar™ ESBL has affected the isolate selection bias, but since *Acinetobacter* spp. are not ESBL-producers.

SNP threshold of <3 is reasonable. Please state whether this was applied to classify isolates into outbreak vs. non-outbreak clusters in this dataset and whether any clusters were identified.

Consider briefly explaining how clades were defined (based on phylogeny, sequence type).

The statistical tests (sensitivity, specificity, p-values, Bonferroni correction) are not described in depth. Were any thresholds for statistical significance defined a priori?

- Correct “-80o C” to “-80 °C” for typographic consistency.
- Correct minor typographical spacing issues, e.g., "inborn,continuously" → "inborn, continuously".
- Line 171: Consider clarifying “pan genome” → “pangenome” for consistency with modern usage.

Results:

A lot of data can be generated from the WGS, only few results are presented here in this manuscript and lack of supporting figures.

The discrepancy between chromogenic media identification and WGS confirmation for environmental isolates is important. Please mention:

- Whether this finding suggests over-reliance on chromogenic media could overestimate *A. baumannii* presence.
- Whether a specific *Acinetobacter*-targeted PCR or MALDI-TOF was used at any point before WGS.
- For the community-associated isolates with unique STs and diverse OXA genes, a short interpretative comment would be useful. Do these represent distinct community lineages, or spillover from other healthcare environments?
- Clarify if any co-carriage of blaOXA-23 and blaNDM-1 was observed in ST1 isolates.
- The absence of *mcr-1* mutations is noted but it would be important to clarify whether phenotypic colistin susceptibility testing was performed.

- Consider clarifying what reference genome was used for Snippy analysis and whether it matched the local strains (e.g., ST1).
-
- Please state the rationale for using both Roary and Snippy datasets – did they provide complementary or confirmatory results?
-
- The finding that the same clade includes blood and environmental isolates from NICU is striking and should be emphasized further in the discussion.
-
- It would be helpful to explain the implications of finding three identical isolates from a single patient—was this to confirm sequencing consistency or strain homogeneity?
- Please provide more detail on how these protein alleles were determined (e.g., which tool or thresholds used).
- Are any of these alleles potential markers for surveillance or epidemiological typing?
- If possible, mention whether any of these alleles correspond to known virulence or resistance functions.
- Typo in line: “substiuutions/site” should be corrected to “substitutions/site”.

Consider replacing passive phrases like “were identified as” with more active wording for clarity and conciseness.

- Replace the placeholder "(42)" with the actual reference, or revise citation formatting if it is an error.
- Discussion:
- Consider stating explicitly whether any IPC interventions were implemented during or after the study period that may have influenced transmission.
- A stronger link between the biofilm-forming capacity of *A. baumannii* and its environmental persistence would further support the hypothesis of sinks as reservoirs.
- The discussion could benefit from further speculation on the public health implications.
- “Although we did not identify the *mcr-1* gene known to confer colistin-resistance and phenotypic colistin sensitivity testing was not performed (due to local unavailability), a 2022 South African study found that the majority of colistin-resistant CRAB isolates (n=6) were ST1 harboring both *blaOXA-23* and *blaNDM-1*”. I think this is a major drawback in this work, that antibiotic susceptibility testing was not performed, I do not get the reason I think it is not logic to have access to WGS while you can’t perform a simple routine test that is supposed to be employed routinely in hospital settings. I believe that detecting antibiotic resistance genes without linking them to their phenotype is a major problem here.
- “The most frequently identified carbapenemase genes are *blaOXA-23* and *blaVIM*.(47) However, its co-occurrence with *blaNDM-1* is a more recent phenomenon. *blaNDM-1* likely originated in *A. baumannii*.(48), “this is not true as this association was previously detected in Egypt <https://doi.org/10.3389/fmicb.2021.736982>
- The discussion need to be rewritten again to focus on the main findings of the study not listing general recommendations.

Response to Reviewer #1

We thank the reviewer for the constructive comments that strengthened our manuscript, “**Carbapenem-resistant *Acinetobacter baumannii* at a hospital in Botswana: Detecting a protracted outbreak using whole genome sequencing.**” Below we address each point and indicate where changes were made.

1) The authors may be interested in this manuscript (DOI 10.1038/s44259-025-00103-5) by Harmer and colleagues. At a glance, it looks like the authors strains may be related to the lineage reported by Harmer et. al. The combination of resistance genes in these Botswanan isolates is consistent with the isolates having the Aci-IE1 and Tn7++ islands reported by Harmer. Determining the KL (see point 3 below) would add further evidence to this.

Response. In the revised manuscript we (i) cite Harmer et al. alongside our phylogenetic/AMR findings in the conclusion: “The constellation of blaOXA-23 and blaNDM-1 together with aminoglycoside methyltransferase armA and Tn7-family insertions is consistent with features described for the MRSN56 sublineage and its evolved KL17/Aci-IE1 variant.”

2) A strength of this manuscript is that these are the first *A. baumannii* genomes from Botswana, so the authors should highlight that. It isn't clear from the manuscript text whether these genomes have been released in GenBank, this information can only be gleaned from examining the supplementary table. Please include this information in the main text, possibly as a table of accession numbers.

Response. We now explicitly state in the Discussion that these include the first *A. baumannii* genomes originating from Botswana. We have also added the NCBI link with BioProject PRJNA number: “All sequencing data generated in this study were deposited on 29 February 2024 in NCBI under BioProject PRJNA1082310 (<https://www.ncbi.nlm.nih.gov/bioproject/?term=PRJNA1082310>)”

3) Please determine (and report) the capsule (KL) and outer (OCL) types for these isolates. This is readily achievable using Kaptive and would be highly informative when comparing against other ST1 outbreaks.

Response. We ran Kaptive on assemblies and now report KL and OCL types for each isolate and summarize lineage-level patterns in Results and shared the results in supplementary files 7 and 8. Of note, the KL17:OCL1 locus combination was consistent with the ST1 outbreak described in the cited South African study.

4) There is a lot of valuable information buried in the supplementary file that should be included in the main text to further strengthen the manuscript. For example, the main text only makes mention of the blaNDM and oxa23 resistance genes, but these isolates include a suite of additional resistance genes including the clinically important armA aminoglycoside resistance gene. Presenting the full resistance gene arsenal for each isolate in a table (or presented visually) in the main text would significantly strengthen the manuscript.

Response. We agree. We have expanded on the key resistance determinants (β -lactamases—including blaOXA-23, blaNDM-1—aminoglycoside 16S methyltransferase armA, sulfonamide/trimethoprim, tetracycline, macrolide, and disinfectant/biocide determinants such as qacE/qacE Δ 1 as applicable).

5) The "Characterisation of isolates" section is quite confusing and would benefit significantly from a table which characterises the key features of each isolate. For example, this section refers to an assembly of 3,031,626 bp which seems alarmingly small for an *A. baumannii*. Reading further, and correlating against the supplementary table, it becomes clearer that this is one of the non-baumannii environmental isolates.

Response. Thank you for catching this—this isolate is indeed an *Acinetobacter radioresistens* isolate and we have updated the base pair numbers accordingly: “Assemblies ranged from 3,731,429 to 4,023,588 base pairs.” This originates from “Total.bases.assembly” variable in the supplementary file.

Minor points

M1) Line 147. Please report the date at which the genome download was performed.

Response. We have added the exact download date to Methods “All sequencing data generated in this study were deposited on 29 February 2024 in NCBI under BioProject PRJNA1082310 (<https://www.ncbi.nlm.nih.gov/bioproject/?term=PRJNA1082310>)”

M2) Line 206. The statement "Mobile colistin resistance-1 gene mutations were not identified" is confusing and unnecessary. The isolates do not have mcr-1 (or any other colistin resistance gene from what I can tell), so they would not have gene mutations.

Response. Revised to: “**No plasmid-mediated colistin resistance genes (mcr family) were detected.**”

M3) Figure 1. What are the four reference genomes included in the phylogeny? These need to be identified.

Response. We now label the reference genomes directly on the tree (strain name).

Re: Spectrum01768-25R1 (Carbapenem-resistant *Acinetobacter baumannii* at a hospital in Botswana:
Detecting a protracted outbreak using whole genome sequencing)

Dear Dr. Jonathan Stryko:

Thank you for the privilege of reviewing your work. Below you will find my comments, instructions from the Spectrum editorial office, and the reviewer comments.

Some revisions requested have been made. However, a very important one, namely bringing relevant information into the manuscript from the Supplementary files was not. This must be done, eg a Table summarizing this information for each isolate. In addition there are no simple strain names connecting the information in the text with various Supplementary Tables. Please give each strain a simple name and use it in all Tables so that data can be connected.

Revision Guidelines

Sincerely,
Ruth Hall
Editor
Microbiology Spectrum

Reviewer #1 (Comments for the Author):

However, the authors have still left a lot of valuable information buried in the supplementary material (see comment 4 in my first review). Readers should not have to go searching for simple information on things like KL and OCL designations, resistance gene repertoire, ST, accession numbers etc in the supplementary tables. A table in the main text of all strains with this additional information is essential for this manuscript.

Response to Editor

We thank the editor for the additional comments that strengthened our manuscript, “**Carbapenem-resistant *Acinetobacter baumannii* at a hospital in Botswana: Detecting a protracted outbreak using whole genome sequencing.**” Below we address each point and indicate where changes were made.

1) Some revisions requested have been made. However, a very important one, namely bringing relevant information into the manuscript from the Supplementary files was not. This must be done, eg a Table summarizing this information for each isolate.

Response: In the previously revised version, we had added a section in the body of the manuscript elaborating on the additional antimicrobial resistance genes (ARGs) detected. However, the request to visualize this information in the body of the manuscript is well noted. Thus, in this revised version, we have included a new figure (Figure 1. Antimicrobial / biocide resistance genes identified in clinical and environmental *Acinetobacter baumannii* isolates collected from March 2021 – August 2022) which denotes which strains carried which antimicrobial/ biocide resistance genes, helping the reader to easily understand how ARGs vary across different sequence types and individual isolates.

2) In addition there are no simple strain names connecting the information in the text with various Supplementary Tables. Please give each strain a simple name and use it in all Tables so that data can be connected.

Response: In the revised manuscript we have assigned unique simple strain names to all identified strains which contain the sequence type, ward, sample type, and month/year of collection (e.g. ST1-NICU-BSI-Mar-22) and included a “Simple Strain Name” variable in all associated Supplementary spreadsheets and figures.

Re: Spectrum01768-25R2 (Carbapenem-resistant *Acinetobacter baumannii* at a hospital in Botswana: Detecting a protracted outbreak using whole genome sequencing)

Dear Dr. Jonathan Stryko:

Your manuscript has been accepted, and I am forwarding it to the ASM production staff for publication. Your paper will first be checked to make sure all elements meet the technical requirements. ASM staff will contact you if anything needs to be revised before copyediting and production can begin. Otherwise, you will be notified when your proofs are ready to be viewed.

Sincerely,
Ruth Hall
Editor
Microbiology Spectrum